# Simulated Microgravity Effects on Human Adenocarcinoma Alveolar Epithelial Cells: Characterization of Morphological, Functional, and Epigenetic Parameters

**DOI:** 10.3390/ijms22136951

**Published:** 2021-06-28

**Authors:** Paolo Degan, Katia Cortese, Alessandra Pulliero, Silvia Bruno, Maria Cristina Gagliani, Matteo Congiu, Alberto Izzotti

**Affiliations:** 1UO Mutagenesis and Preventive Oncology, IRCCS Ospedale Policlinico San Martino, 16132 Genoa, Italy; paolo.degan@virgilio.it; 2Cellular Electron Microscopy Laboratory, Department of Experimental Medicine, University of Genoa, 16132 Genoa, Italy; cortesek@unige.it (K.C.); gagliani@unige.it (M.C.G.); 3Department of Health Sciences, Department of Health Sciences University of Genoa, 16132 Genoa, Italy; alessandra.pulliero@unige.it; 4Department of Experimental Medicine, University of Genoa, 16132 Genoa, Italy; silvia.bruno@unige.it (S.B.); S3370203@studenti.unige.it (M.C.)

**Keywords:** microgravity, A549 cell, microRNA, electron microscopy, mitochondria, lung cancer

## Abstract

Background: In space, the reduction or loss of the gravity vector greatly affects the interaction between cells. Since the beginning of the space age, microgravity has been identified as an informative tool in biomedicine, including cancer research. The A549 cell line is a hypotriploid human alveolar basal epithelial cell line widely used as a model for lung adenocarcinoma. Microgravity has been reported to interfere with mitochondrial activity, energy metabolism, cell vitality and proliferation, chemosensitivity, invasion and morphology of cells and organelles in various biological systems. Concerning lung cancer, several studies have reported the ability of microgravity to modulate the carcinogenic and metastatic process. To investigate these processes, A549 cells were exposed to simulated microgravity (µG) for different time points. Methods: We performed cell cycle and proliferation assays, ultrastructural analysis of mitochondria architecture, as well as a global analysis of miRNA modulated under µG conditions. Results: The exposure of A549 cells to microgravity is accompanied by the generation of polynucleated cells, cell cycle imbalance, growth inhibition, and gross morphological abnormalities, the most evident are highly damaged mitochondria. Global miRNA analysis defined a pool of miRNAs associated with µG solicitation mainly involved in cell cycle regulation, apoptosis, and stress response. To our knowledge, this is the first global miRNA analysis of A549 exposed to microgravity reported. Despite these results, it is not possible to draw any conclusion concerning the ability of µG to interfere with the cancerogenic or the metastatic processes in A549 cells. Conclusions: Our results provide evidence that mitochondria are strongly sensitive to µG. We suggest that mitochondria damage might in turn trigger miRNA modulation related to cell cycle imbalance.

## 1. Introduction

Gravitational modulations have a great impact on living organisms as all forms of life have evolved from the very beginning under the constant of gravity force G. In space, the reduction or loss of the gravity vector greatly reduces friction, convection, and hydrostatic forces and this condition affects many fundamental functions at the organism, organ, tissue, and cellular level [1]. The modulation of gravity should therefore be considered a powerful tool in the investigation of important physiological processes.

As microgravity disturbs the interaction between cells and their environment, changes are seen at the cellular and sub-cellular levels involving morphological, immunological, biochemical, metabolic, transduction, and functional changes. Since the beginning of the space-era, back on the past middle century, microgravity has been regarded as an unconventional but extremely informative tool in biology and medicine including cancer research [2].

Since real microgravity can only be attained in orbital laboratories, an unpractical and expensive condition, several dedicated devices which allow to simulate the loss of the gravity vector have been developed. Simulated microgravity (µG) is commonly attained in laboratory conditions with a random position machine (RPM) [3]. 

Lung cancer is a leading cause of cancer-related mortality worldwide accounting for 11.4% of total new diagnosed cancer cases and accounting for the 18% of total cancer deaths in 2020 [4]. Lung cancer is classified in two main types: small-cell (SCLC) and non-small-cell lung carcinoma (NSCLC). SCLC accounts for around 12–15% of all cases and is more aggressive and metastatic than NSCLC, less aggressive and slow with respect to the SCLC but more common and accounting for more than the 85% of all lung cancer cases. NSCLC is classified in three subtypes: adenocarcinoma (50%), squamous cell carcinoma (30%), and large cell carcinoma (10%). These subtypes start from different types of lung cells and are grouped together because their treatment and prognoses are similar [5]. Among human lung cancer derived cell lines, the lung adenocarcinoma cell line A549, established in 1972 by D.J. Giard (ATCC cell line bank CCL 185TM), is an hypotriploid human alveolar basal epithelial cell line widely used as an in vitro model for type II pulmonary epithelial cells and a model for lung adenocarcinoma [6]. A549 cells grow in a monolayer in standard culture laboratory conditions.

µG has been reported to interfere with mitochondrial activity, energetic metabolism, cell viability and proliferation, chemosensitivity, invasion and cell and organelle morphology in different biological systems (glioblastoma U87 cells [7]; malignant glioma [8]; normal and cancerous breast cells [9]; muscle cell [10]; lymphoblasts [11]. These studies evidence how cell types and lineages are differently affected by µG and several other factors not yet completely understood may concur to the final behavior. 

Concerning lung cancer several studies reported the ability of µG to modulate the carcinogenic and metastatic process [12,13,14] and despite the many important results obtained, no conclusive conclusion can be drawn to date.

Our approach was meant to define some aspects of the exposure of A549 to µG in relationship with cell cycle and structural defects. Furthermore, we also performed a global analysis of A549 miRNAs exposed for different time to µG. This approach allowed to define a pool of miRNA associated with this solicitation. To our knowledge, this is the first global miRNA analysis of µG exposed A549 reported. We are confident that this study will provide useful information for further studies to better define the role of µG in the carcinogenic process.

## 2. Results

### 2.1. Cell Proliferation and Cell Cycle

µG significantly affects A549 cells proliferation and their cell cycle. Table 1 reports data associated to cell proliferation and cell cycle for A549 cells maintained in simulated µG and normal gravity (NG) conditions for 24, 36, and 48 h. Exposure of A549 cells to µG leads to a reduction of cell proliferation, associated with a progressive accumulation in the G1 and G2 cell cycle phases. The morphological parameters of A549 maintained in µG are essentially stable in the range between 0 and 48 h. The apoptosis was measured by Trypan blue assay.

### 2.2. µG Induces the Formation of Polynucleated A549 Cells

As reported on Figure 1A, µG exposure of A549 cells is accompanied by the generation of large polynucleated cells in comparison with ground cells. This process is observed with a frequency between 10 and 15%, with cells displaying multinucleated cells (in the range from 2 to 8 nuclei, red arrowheads) at 24 h exposure. In addition, nuclei of µG exposed cells are significantly larger than in untreated cells (Figure 1B). In longer exposures, the proportion of multinucleated cells is maintained at approximately the same level probably because of low cell proliferation (see Table 1 and Figure 2).

### 2.3. Transmission Electron Microscopy

To better characterize the alterations occurring in A549 cells exposed to 24 and 36 h of µG, we analyzed mitochondria morphology at the ultrastructural level by transmission electron microscopy.

As show in Figure 3A,B, in NG, mitochondria display normal cristae and electron-density. Abundant endoplasmic reticulum (ER) cisternae are also visible surrounding them. In contrast, 24 and 36 h of µG strongly alter mitochondria morphology resulting in abnormal and dilated cristae with increased electron-density. ER cisternae often appear dilated. Notably, several autophagic/auto lysosomal vacuoles are also evident at 24 and 36 h, which are rarely observed in ground control cells. This latter observation suggests the possible activation of a degradative mechanism that dispose of damaged organelles.

### 2.4. MicroRNA Analysis of A549 Cells in NG versus µG Exposure at 36 Hours

Since cell parameters in the interval 24–48 h is essentially stable, we choose to perform microRNA analysis of cells at 36 h. The microRNAs displaying a two-fold up- or down-regulation (>2-fold variation and *p* < 0.05) were selected (Appendix A). A total of 26 down-regulated and 32 up-regulated miRNAs have been selected.

### 2.5. Enrichment Analysis Strategy

Analysis of up- and down-regulated microRNAs were accomplished according to Hong and Stoney [15,16]. A549 µG/NG microRNAs screening analyzed with DIANA Tools provided a heat map to enrich the miRNA selected (Appendix A). According to this enrichment procedure, 19/26 of the down-regulated and 17/32 of the up-regulated miRNAs of the µG/NG were significant in terms of functional associations. In order to increase the stringency of this approach, the ‘signature’ obtained with DIANA analysis was subsequently processed with two more software’s (TAM2.0 and Mienturnet) (Table 2).

These procedures finally result in the selection of the most representative gene population associated with the up- and down-regulated microRNAs. Thirteen up-regulated, 38 down-regulated genes were identified. Six genes are shared among these two populations.

The graphic representation of the gene set reported on Figure 4 with STRING also defined the molecular and biochemical relationship relevant to the exposure of the A549 cells to µG.

MiRNAs modulated by µG are mainly involved in cell cycle regulation (P53, CDKN, E2F), apoptosis (BCL2, BIRC5), stress response (FOS, MAPK).

## 3. Discussion

In the present study, we reported the effect of µG exposure in the A549 cells. Cells continuously exposed to µG display growth inhibition, cell cycle unbalance, apoptosis, and gross morphological abnormalities including damaged mitochondria and the presence of autophagic vacuoles. Furthermore, the exposure of A549 cells to µG is accompanied by the generation of polynucleated cells suggesting profound deregulation of cell cycle and DNA replication. Concerning the cell cycle unbalance and polynucleation in A549 cells, two recent articles commented on these points. The fist paper [17] dealt with the generation of A549-polyploid giant cancer cells (PGCCs) after Staurosporine (STS) treatment. In this context, the authors commented that polynucleated A549 cells might function as blastomere-like stem cells and thus play some roles in tumor heterogeneity, stemness, and resistance. In the second article [18], two human lung cancer cells lines, A549 and H1299, were treated with reversine acting as an anticancer agent able to suppress the proliferation of multiple human cancer cells, through actions such as cell cycle arrest, apoptotsis, and autophagy induction.

According to these observations, the induction of polynucleation in A549 appears as a potential protective and anticarcinogenic effect. Thus, µG exposure might be regarded as a tumor suppressing condition. This finding appears in line with several recent observations [19,20]. These papers have reviewed the effect of microgravity and how it could influence the course of cancer development.

In this article, we reported that exposure of A549 cells to µG is associated with several relevant morphological and physiological changes. To establish how these changes are related to the cell’s biochemistry and metabolism, we undertook a differential analysis of microRNA expression in µG vs. NG.

Based on these results, we focused our attention on a detailed microRNA analysis in A549 cells exposed to normal gravity or µG to highlight a potential signature that would define the µG condition. It is known that the relationship between miRNAs and mRNA is degenerated. Indeed, a single miRNA binds many mRNA sequences and, vice-versa, each mRNA sequence is target of many miRNAs. Bioinformatics offers powerful tools to gather massive information concerning microRNAs relationships with molecular targets, biochemical pathways, biological processes and functional relationships, and disease ontology. In this wealth, however, it is difficult to find a pathway that allows the generation of crucial and sound information.

In this study, the analysis reported is consistent with handling the pool of miRNAs selected in the enrichment process as a signature. In-silico approaches commonly applied in mining informatics data are strongly biased since they lead to the identification of highly related biological processes [16]. To avoid this problem, we adopted approaches meant to avoid over-counting miRNAs targeting multiple genes in the same pathway [16,21]. As described above, the enrichment analysis was performed employing the miRNA signature obtained after the highest stringency treatment. The microRNAs reported in Table 2 and the differentially expressed genes related to these microRNAs quoted on Figure 4 represent the activities selectively altered by the µG exposure of A549 cells. In this article, we demonstrated that A549 cell exposure to µG is associated with several biochemical and molecular modifications linked to a series of pathological processes.

A relatively small number of genes appear responsible for the characteristic modifications in A549 exposed to µG.

These genes are closely related in central metabolic pathways where the phosphatidylinositol 3-kinase-Akt (PIK3R1-Akt) signaling pathway plays a crucial role [22] PIK3R1-Akt is primarily regulated upstream with their negative regulator PTEN and downstream with several effectors including FOXO3. On another side, these genes are linked with cell cycle control (CDKN2A/1A and RB) and with the transcription factors E2F1/3 and, finally with the control of the mitochondrial activity (BCL2). The control of all these highly interconnected paths appears under the control of TP53. The STRING representation (Figure 4) of these relationships offers a picture of this complex pattern.

In conclusion, the relationships sketched in Figure 4 may be used as tools in the experimental activity meant to define the µG biological effects.

To our best of our knowledge, this is the first study that reported a microRNA analysis on A549 cells exposed either to real or µG. Previous studies that associate miRNA analyses in A549 and NSCLC cells have been published in recent years. However, many of these studies are focused on the evaluation of the role of single miRNAs on A549 phenotype. A study [23] reported the variations in the global miRNA expression profiles from high and low invasive NSCLC cell lines (A549 and SK-Lu1). Further, 11 miRNAs were associated with NSCLC metastatic potential. Other studies [24,25] examined in more detail the association between microRNAs and early detection of non-small cell lung cancer (NSCLC). We found no correlation between the microRNA population related to the pathological NSCLC process and the population of microRNA modulate by µG.

## 4. Materials and Methods

### 4.1. Cells and Reagents

A549 cells were maintained in culture in RPMI 1640 medium supplemented with 10% FCS, 25 mM Hepes, and 2 mM L-Glutamine at 37 °C at 5% CO_2_. Cells were grown in 20 × 20 mm glass coverslips. Cell density was 1 × 10^5^ cells per coverslip. In the various experiment reported below, cells were grown, treated, and analyzed under identical conditions except for the absence or presence of microgravity. Cell proliferation was quantified through the determination of the cumulative population doubling level (PDL) calculated as log2 (D/D0) where D is the density of cells when harvesting and D0 is the density of cells when seeding. Cell viability was determined with the Trypan blue dye exclusion test. Cell cycle analysis was accomplished with FACSCalibur (Beckton Dickinson) and with CyAn ADP cytometer (Beckman Coulter). Cells were stained with propidium iodide, and 20,000 events were collected from each sample before ModFit analysis.

### 4.2. Simulated Microgravity (µG)

Simulated microgravity was accomplished by a random position machine (RPM) machine (Dutch Space, Leiden, NL, USA) located in a temperature-controlled room. RPM [1] is a laboratory instrument designed to randomly change the position of an accommodated biological experiment in 3-dimensional space. The lay-out of the RPM consists of two cardanic frames and one experiment platform. The frames and the platform are driven by means of belts and two electro-motors. The RPM is computer managed and a dedicated software permits the settings for modeled microgravity at the value of choice. Rotation rate ω and geometrical distance from the center of rotation (R) yield ‘g-contours’, through g_i_ = ω^2^R/g_0_ (g_0_ = 9.81 m/s^2^), that provide guidelines for the design and lay-out of experiment packages and for the interpretation of the experimental results. Working conditions employed in our experiments sets g below 0.005 m/s^2^. In the conditions employed in the experiments reported below, cells were exposed continuously in the RPM for 24, 36, and 48 h.

### 4.3. Fluorescence Microscopy

Cells were seeded on coverslips and allowed to attach for 24 h. Cells were exposed to µG or maintained in standard conditions (NG) for 24 and 36 h. At the end of treatments, slides were washed with PBS, fixed in methanol, and incubated with DiOC6 for 45 min at 37 °C. Cells were then counterstained with DAPI. Photographs were taken with a Provis AX70 microscope (Olympus, Tokyo, Japan) and Cytovision software (Applied Imaging Corp., Santa Clara, CA, USA.) [26].

### 4.4. Transmission Electron Microscopy

A549 cells were washed out twice in 0.1 M cacodylate buffer and fixed in 0.1 M cacodylate buffer containing 2.5% glutaraldehyde (Electron Microscopy Science, Hatfield, PA, USA), for 1 h at room temperature. The cells were postfixed in 1% osmium tetroxide for 1 h and 1% aqueous uranyl acetate for 1 h. Subsequently, samples were dehydrated through a graded ethanol series and embedded in resin (Poly-Bed; Polysciences, Inc., Warrington, PA, USA) for 24 h at 60 °C. Ultrathin sections (60 nm) were cut and counterstained with 5% uranyl acetate in 50% ethanol. Electron micrographs were acquired at Hitachi 7800 120Kv electron microscope (Hitachi, Tokyo, Japan) equipped with a Megaview G3 digital camera and Radius software (EMSIS, Munich, Germany). For ultrastructural analysis, 30 randomly taken micrographs for each treatment were examined for mitochondria morphology. The results represent the mean of two independent experiments.

### 4.5. RNA Extraction and Analysis/Data Analysis/Validation by Real-Time qPCR

Total RNA from A549 cells was extracted and RNA quantity and quality were analyzed (Nanodrop, ND-1000; Scientific Thermofisher, Wilmington, DE, USA) by calculating 260/230 and 260/280 absorbance ratios. The Qubit quantitation assay was used as performed in Qubit 3.0 Fluorometer (Life Technologies, Gent, Belgium). Sample labeling and miRNA array hybridization. For evaluating the expression of miRNAs, we use the seventh generation miRCURY LNATM microRNA Array (Exiqon, Vedbaek, Denmark), which contains 3100 capture probes covering human, mouse, and rat miRNAs. This microarray analyzes the expression of 1928 human miRNAs. One microgram RNA from each sample was labeled with Label ITmiRNA Labeling Kits, Version 2 (Mirus Bio, Madison, WI, USA), following the standard protocol. Total RNA was mixed with 10 mL 10 labeling buffer, 4 mL Label IT reagent (containing Cy 3 or Cy 5 fluorescent tracers), and water to 86 mL. The samples were incubated at 36 °C for 1 h and the reaction was stopped by adding 10 mL Stop Reagent. Labeled samples were purified on a chromatographic column and eluted in 25 mL elution buffer. Then, Hybridization Solution (EXIQON, Vedbaek, Denmark) was added and the resulting mixture denatured at 65 °C for 3 min. The labeled mix was transferred to the microarray and covered with coverslips. The hybridization was performed in GlassArray Hybridization Cassettes (Invitrogen Ltd., Paisley, UK) in a water bath at 37 °C for 16 h and a wash sequence was performed. The array was dried by centrifugation and laser scanned (ScanArray; PerkinElmer, Waltham, MA, USA) to record fluorescent signals produced by each spotted probe effectively hybridized with the corresponding miRNA.

The microarray data were processed by GeneSpring software. The local background was subtracted, data log-transformed, and normalized per gene/chip mean. Data overall variability, as related to A549 experimental conditions, was examined by box plot analysis, scatter plot analysis, hierarchical cluster analysis, and principal component analysis. Individual miRNAs modulated in the A549 experimental conditions adopted were identified by volcano plot analyses.

Microarray results for miR-16 and miR-34a were further validated by qPCR on A549 cells (Appendix A). These miRNAs were selected because of their relevance in microgravity conditions. Total RNA (10 ng) was reverse transcribed using miR-specific stem-loop RT primers (TaqMan MicroRNA Assays; Applied Biosystems, Thermo-Fisher, Waltham, MA, USA) and components of the High-Capacity cDNA Reverse Transcription kit (Life Technologies, Carlsbad, CA, USA) according to the manufacturer’s protocols. Expression levels of individual miRNAs were detected by subsequent RQ-PCR using TaqMan MicroRNA assays (Life Technologies, Carlsbad, CA, USA) and a Rotor Gene 3000 PCR System (Corbett Research, Mortlake, Australia) (Qiagen) using standard thermal cycling conditions in accordance with manufacturer recommendations. PCR reactions were performed in triplicate in final volumes of 30 µL, including inter-assay controls (IAC) to account for variations between runs. RT-PCR (TaqMan MicroRNA Assays; Applied Biosystems, Thermo-Fisher, Waltham, MA, USA) was used to quantify the expression of miR-16 and miR-34a according to the manufacturer’s instructions. To normalize the data for quantifying miRNAs, the universal small nuclear RNU38B (RNU38B Assay ID 001004; Applied Biosystems) as an endogenous control was used.

### 4.6. Bioinformatic Analysis and Web Tools

The purpose of the use of different and complementary web tools was to get functional enrichment analysis and identify classes of genes or proteins over-represented in the large set of genes associated with the A549 miRNAs population and to associate them with the relevant disease phenotype.

DIANA tools [27] provides algorithms, databases, and software for interpreting and archiving data in a systematic framework ranging from the analysis of expression regulation from deep sequencing data, the annotation of miRNA regulatory elements and targets to the interpretation of the role of ncRNAs in various diseases and pathways. A selection of the most relevant miRNAs associated with the DIANA analysis is obtained through the generation of heat maps which enable the identification of miRNA subclasses or the GO terms that characterize these miRNAs. The original FA-miRNA signature was successively analyzed with MIENTURNET [28], a software which offers the possibility to perform statistical and network-based analyses using an effective tool leading to a more effective prioritization of the miRNA–target interactions. This has the potential to allow researchers with limited computational and informatics experience to gather information and exhaustively thanks to the intuitive web interface. TAM, now in its updated version TAM2.0 [29], is a software for miRNA set enrichment analysis. This resource is a manual curation which covers 9945 and 1584 newly collected miRNA-disease and miRNA function associations, respectively. TAM2.0 allows to test the functional and disease annotations of miRNAs by overrepresentation analysis and compare the input deregulated miRNAs with those de-regulated in other disease conditions via correlation analysis. GSEA (Gene Set Enrichment Analysis) [30,31] evaluates microarray data at the level of gene sets. These gene sets are based on prior biological knowledge, published information about biochemical pathways or experimental data. Then, GSEA is meant to correlate a given set of genes grouped in the same biological pathway or by proximal location on a chromosome. GSEA represents one of the latest approaches to gene expression analysis. This method uses aggregated gene sets to identify biological processes present across phenotypes in microarray data sets. This approach greatly facilitates understanding the underlying biology driving selected pathological phenotypes and facilitate the molecular characterization of human disease. In the end, also STRING (Search Tool for the Retrieval of Interacting Genes/Proteins), a biological database and web resource of known and predicted protein–protein interactions [32] allowed a graphical representation of the gene sets associated with A549 µG exposure.

## 5. Conclusions

This is the first study which reports a global analysis of microRNA associated to µG exposure in A549 cells.

Our experimental findings provide evidence that mitochondria are very sensitive to µG as well detectable by their morphological alterations at TEM. This situation in turn reflects onto the microRNA machinery and cell cycle. The microRNAs highlighted in this study do not appear to correlate with the NSCLC pathological progress. They might instead be regarded as potential therapeutics of the condition associated to µG. The implications of this finding deserve further scrutiny.

## Figures and Tables

**Figure 1 ijms-22-06951-f001:**
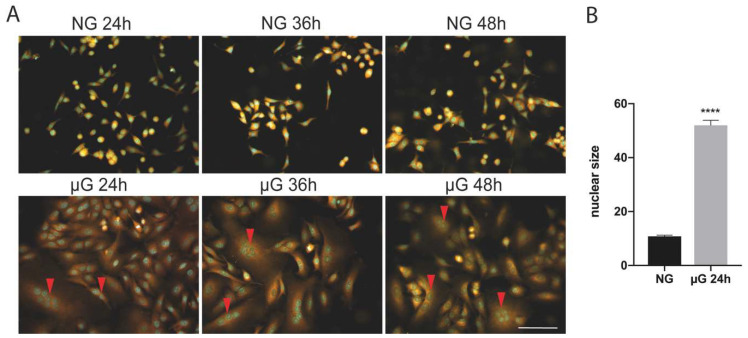
(**A**) Microgravity exposure of A549 cells is accompanied by the generation of polynucleated cells at 24 and 36 h. Multinucleated cells were observed during the various exposure times at high frequency (10–15% of the total). Scale bar: 100 µm. (**B**) Histogram showing the nuclear size (mean+/−SEM) in normal gravity (NG) versus 24 µG exposed cells (*n* = 100 cells, **** *p* < 0.0001, T Student’s test).

**Figure 2 ijms-22-06951-f002:**
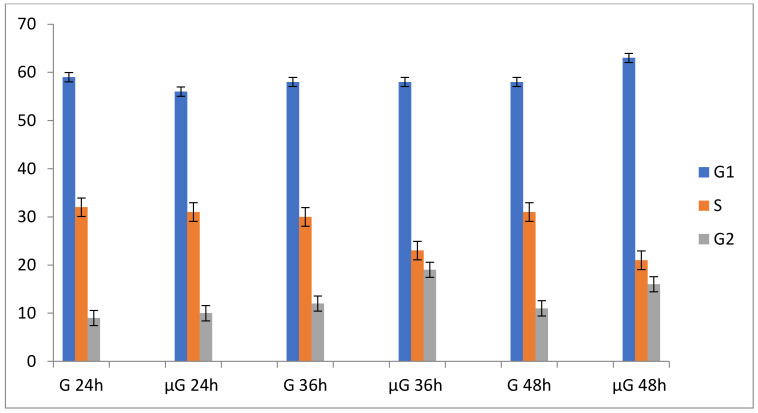
Time course fluctuations in population doubling (PD) and cell cycle phases G1, S, and G2 of A549 cells in normal gravity (G) and in simulated microgravity (µG).

**Figure 3 ijms-22-06951-f003:**
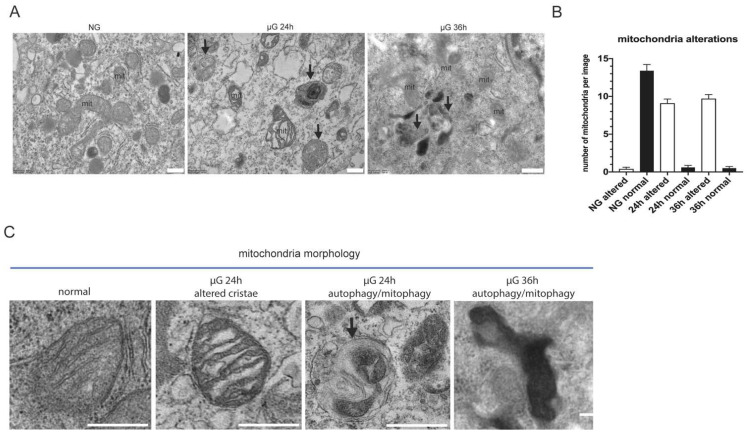
Ultrastructural analysis of mitochondria morphology under µG. Representative micrographs of A549 cells subjected at normal versus µG for 24 and 36 h. (**A**) Normal gravity mitochondria (mit) display normal cristae and electron-density. Abundant endoplasmic reticulum (ER) cisternae are visible surrounding them. Altered mitochondria (mit) with abnormal cristae and increased electron-density are observed in µG-treated cells. ER cisternae appear dilated. Several autophagic/auto lysosomal vacuoles are evident at both 24 and 36 h (black arrows), and are rarely observed in ground cells. Scale bars: 500 nm. (**B**) Histogram showing the number of altered mitochondria per micrograph in each experimental condition (mean+/−SEM). (**C**) High magnification details of mitochondria morphology alterations in µG compared to mitochondria in normal gravity. Note the alterations of cristae, the increased electron-density, and the presence of an autophagic vacuole containing a putative damaged mitochondrion (black arrow). Scale bars: 500 nm.

**Figure 4 ijms-22-06951-f004:**
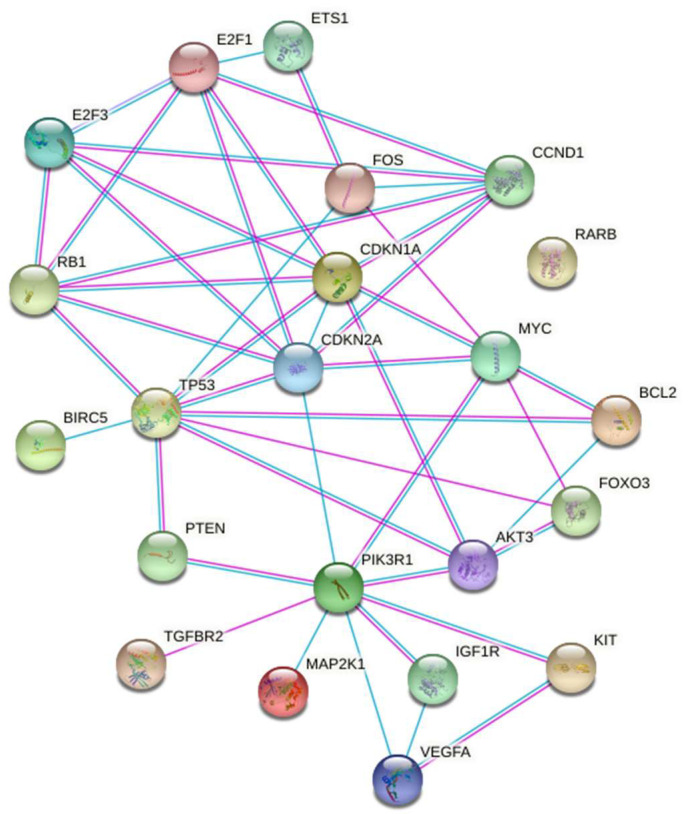
STRING representation of the molecular and biochemical relationships among the genes selected after the enrichment analysis.

**Table 1 ijms-22-06951-t001:** Time course fluctuations in population doubling (PD) and cell cycle phases of A549 cells in NG and µG.

	Time (Hours)	PD	G1	S	G2	% Trypan Blue Stained Cells
Ground (G)	24	1.00	59 ± 3	32 ± 3	9 ± 1	2 ± 2
	36	1.43 ± 0.32	58 ± 1	30 ± 3	12 ± 2	3 ± 1
	48	1.72 ± 0.27	58 ± 2	31 ± 2	11 ± 3	4 ± 3
Simulated Microgravity u(g)	24	1.00	56 ± 2	31 ± 3	10 ± 2	3 ± 3
	36	0.96 ± 0.22 **	58 ± 3	23 ± 4 **	19 ± 3 **	4 ± 5
	48	1.12 ± 0.18 **	63 ± 5 *	21 ± 5 **	16 ± 3 *	6 ± 5

Data reported on Table 1 were from 4 replicates. PD was calculated from cells growing on flasks at an initial density of 10 × 10^5^ cells/cm^2^. Cell cycle data were obtained after collection of 20,000 events in cytometry for each sample. Trypan blue data were obtained after staining cells grown on coverslip. (* *p* < 0.05; ** *p* < 0.005).

**Table 2 ijms-22-06951-t002:** microRNAs down- and up-regulated in A549 cells selected after enrichment procedures.

DOWN	UP
hsa-miR-16-5p	hsa-let-7b-5p
hsa-miR-194-5p	hsa-miR-107
hsa-miR-20a-5p	hsa-miR-193b-3p
hsa-miR-221-3p	hsa-miR-29a-3p
hsa-miR-30c-5p	
hsa-miR-34a-5p	

## Data Availability

Research Data are available on request to the PI.

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
