# Peer review of "Simulated Microgravity Effects on Human Adenocarcinoma Alveolar Epithelial Cells: Characterization of Morphological, Functional, and Epigenetic Parameters"

_ijms, 2021, doi:10.3390/ijms22136951_

Round 1
Reviewer 1 Report
The article’s intent is to characterise the changes of morphology, function and epigenetics of human adenocarcinoma alveola epithelial cells in response to microgravity.
The results presented do not provide sufficient clarity in these processes, nor explain the importance of any changes in the context of lung cancer.
If this is the first miRNA analysis of A549 cells in ug, i would expect much more detailed results and analyses, and conclusions drawn regarding other analysis performed herein.
The authors introduce small-cell and non-small-cell lung cancer and the A549 cell line that appears (though is not clear) to represent a non-small-cell cancer, but the study does not sufficiently address the carcinogenic or metastatic processes that may be altered by microgravity in this cell, despite alluding to it in the introduction.
Regarding the overall language – there are numerous changes that are required but I am not going to list them here. Also, microgravity should be (µg) as an acronym, not MG?
The results are limited in scope.
2.1.1
The cell proliferation/cell cycle result do not tell us any statistical significance. There is no indication as to the number of cells analysed. It make more sense to show this table graphically, particularly since the figure legend appears to also include normal cells in normal conditions. The methods lack detail as to how these cells were cultured – flasks, amount of media, and so on.
2.1.2
Polynucleated cells. The authors state that the process is observed at ‘very high frequency’. This is a qualitative result. The “10-15%” of cells is not detailed numerically – how many cells were analysed? How were they analysed? How many replicates?
There is no discussion as to the importance, if any, of multinucleated cells and what this might mean for lung cancer. What are the potential reasons for multinucleation.
What happens to multinucleated cells when re-introduced to 1 g ?
There are no scale bars on figure 1. There is a lack of supplementary data – more images are required that sufficiently show these changes. Where are the statistics on cell size, etc? The volume of nuclei to cell volume ratio would be interesting.
2.1.3 TEM.
There is no scale on enlargements Figure 2B. How many cells were analysed? How many images were taken, and what number of altered phenomenon were observed? Surely there should still be autophagic vacuoles observed in normal gravity?
2.1.4. This is not a result that means anything.
What ‘cell parameters’ that are ‘essentially stable’? This isn’t scientific. The reasoning for choosing 36h and not 24h is poor – if micrographs are being shown at 24h, surely results that match this time point should be used. A heatmap to show these miRNAs altered, perhaps even 1.5-fold, in a complete figure, would be useful. Further… how many replicates, cells, and so forth.
2.1.5. Enrichment analysis.
I do not understand, from this result, what is being shown. miRNAs affect multiple genes. But what has the list of other cancers to do with this? Figure 4 shows that all (or some? It isn’t stated). Of the genes are connected. No conclusions are drawn from this. And none in reference to NSCLC. What does enrichment, to produce Table 2, involve? Why are miRNAs that are altered (supp. Data), excluded?
Is this paper focussed on miRNA? In which case, what relevance is population doubling time or multinucleation?
The discussion and conclusions drawn to not adequately explain any results found.
Author Response
Dear Editor:
thank you for your letter stating that our manuscript entitles: “Microgravity effects on human adenocarcinoma alveolar epithelial cells. Characterization of morphological, functional, and epigenetic parameters”, should be acceptable for publication in International Journal of Molecular Sciences pending revisions.
Accordingly, we prepared a revised version of the manuscript acknowledging Referees’ and Editor’s comments as below specified:
Reviewer 1:
COMMENT 1.
The results presented do not provide sufficient clarity in these processes, nor explain the importance of any changes in the context of lung cancer. If this is the first miRNA analysis of A549 cells in ug, i would expect much more detailed results and analyses, and conclusions drawn regarding other analysis performed herein.
ANSWER 1
We thank the reviewer that allowed us to better focus on this point. We have now clearly specified in conclusion (both in the Text and in the Abstract) that ‘Conclusions: as mitochondria represent a major µG sensor inside the cell, our results confirm that mitochondria are strongly sensitive to µG. We suggest that mitochondria damage might in turn trigger miRNA modulation related to cell cycle imbalance.
COMMENT 2
The authors introduce small-cell and non-small-cell lung cancer and the A549 cell line that appears (though is not clear) to represent a non-small-cell cancer, but the study does not sufficiently address the carcinogenic or metastatic processes that may be altered by microgravity in this cell, despite alluding to it in the introduction.
ANSWER 2
We thank the reviewer that allowed us to better focus on this point. We realized that the premises of this paper were not presented with sufficient clarity. To this end we than modify several sentences in the abstract of the manuscript. The changes are highlighted in yellow. The purpose in this article was to compare the effect of simulated microgravity (µG) exposure to A549 cells (thus µG-unexposed cells toward µG-exposed cells). The question whether is µG able to interfere with the carcinogenic or the metastatic processes is an open question. As can be evinced by the sentences at the end of the discussion (lines 193 – 198) our findings are different from those acknowledged in references 18, 19 and 20. Thus, in our hands, simulated µG exposure cannot be ruled out as a pro-metastatic or pro-cancerogenic solicitation.
COMMENT 3.
Regarding the overall language – there are numerous changes that are required but I am not going to list them here. Also, microgravity should be (µg) as an acronym, not MG?
ANSWER 3
The acronym of microgravity on the text has been changed and corrected. We reported µG and not µg, it could be confused with micrograms.
COMMENT 4
The results are limited in scope.
2.1.1 The cell proliferation/cell cycle result do not tell us any statistical significance. There is no indication as to the number of cells analysed. It make more sense to show this table graphically, particularly since the figure legend appears to also include normal cells in normal conditions. The methods lack detail as to how these cells were cultured – flasks, amount of media, and so on.
ANSWER 4
Thank you again to the reviewer for the suggestions. The text in 2.1.1 and 4.1 was amended and more experimental details were included including the number of cells analysed and the conditions of cell growth. Data on Table 1 were modified with introduction of statistics. Data for time 72 exposure were deleted. Time course fluctuations in population doubling (PD), cell cycle and trypan blue data were from 4 times replicated experiments.
COMMENT 5
2.1.2 Polynucleated cells. The authors state that the process is observed at ‘very high frequency’. This is a qualitative result. The “10-15%” of cells is not detailed numerically – how many cells were analysed? How were they analysed? How many replicates?
ANSWER 5
We acknowledge the reviewer for the comments. Images were taken after DAPI and DiOC6 immunohistochemical staining of µG-exposed cells. In each field 2-3 cells among 20 resulted as polynucleated cells. In addition, we added an histogram showing the quantification of the increase in nuclear size in ground and in 24h µG conditions.
COMMENT 6
There is no discussion as to the importance, if any, of multinucleated cells and what this might mean for lung cancer. What are the potential reasons for multinucleation. What happens to multinucleated cells when re-introduced to 1 g ?
ANSWER 6
We acknowledge the reviewer for the comments. Question 1 – To our best knowledge only two papers related with polynucleation in A549 cells.
The fist paper (Glassmann A, Carrillo Garcia C, Janzen V, Kraus D, Veit N, Winter J, Probstmeier R. Staurosporine Induces the Generation of Polyploid Giant Cancer Cells in Non-Small-Cell Lung Carcinoma A549 Cells. Anal Cell Pathol (Amst). 2018 Oct 10;2018:1754085. doi: 10.1155/2018/1754085. eCollection 2018.) deals with the generation of A549-polyploid giant cancer cells (PGCCs) after Staurosporine (STS) treatment. In this context the authors comment that polynucleated A549 cells might function as blastomere-like stem cells and thus play some roles in tumor heterogeneity, stemness, and resistance. In the second article (Lu YC, Lee YR, Liao JD, Lin CY, Chen YY, Chen PT, Tseng YS. Reversine Induced Multinucleated Cells, Cell Apoptosis and Autophagy in Human Non-Small Cell Lung Cancer Cells. PLoS One. 2016 Jul 6;11(7):e0158587. doi: 10.1371/journal.pone.0158587. eCollection 2016) two human lung cancer cells lines, A549 and H1299, were treated with reversine acting as an anticancer agent able to suppress the proliferation of multiple human cancer cells, through actions such as cell cycle arrest, apoptotsis and autophagy induction.
According to these observations the induction of polynucleation in A549 appear as a potential protective and anticarcinogenic effect. Thus, µG exposure might be regarded as a tumor suppressing condition. This finding appears in line with several recent observations (Jhala DV, Kale RK, Singh RP. Microgravity alters cancer growth and progression. Curr Cancer Drug Targets. 2014;14(4):394-406. doi:10.2174/1568009614666140407113633.; Takahashi A, Suzuki H, Omori K, Seki M, Hashizume T, Shimazu T, Ishioka N, Ohnishi T. Expression of p53-regulated proteins in human cultured lymphoblastoid TSCE5 and WTK1 cell lines during spaceflight. J Radiat Res. 2012;53(2):168-75. doi: 10.1269/jrr.11140. Epub 2012 Feb 25.). These papers have reviewed the effect of microgravity and how it could influence the course of cancer development.
The above reported text was included at lines 180-202 in the revised text.
Question 2 – We are sorry to admit that it was not our purpose to explore the fate and the outcomes of the cells that were exposed to microgravity. This issue might be of interest for future experimentations.
COMMENT 7
There are no scale bars on figure 1. There is a lack of supplementary data – more images are required that sufficiently show these changes. Where are the statistics on cell size, etc? The volume of nuclei to cell volume ratio would be interesting.
ANSWER 7
We thank the reviewer for the comments. We now provide a modified Figure 1 A e B that better show the effect of µG on the generation of polynucleated cells. Red arrowheads are added to highlight polynucleated cells. Scale bar is now included. In µG cells, nuclei and the whole cells clearly appear larger with respect with normal gravity cells. We have added Figure 1B, a histogram showing the increased size of the nuclei upon 24h µG. In the legend are reported the statistical significance and number of cells counted.
COMMENT 8
2.1.3 TEM.
There is no scale on enlargements Figure 2B. How many cells were analysed? How many images were taken, and what number of altered phenomenon were observed? Surely there should still be autophagic vacuoles observed in normal gravity?
ANSWER 8
We acknowledge the reviewer for the comments. Figure 2 has been modified by adding a representative image of mitochondria at 36 hours of µG and the quantification of altered mitochondria in each condition (Figure 2A and B). The number of altered mitochondria in µG with respect to ground conditions is now reported as histogram (mean+/- SEM) (Figure 2B). In addition, we have now added scale bars in the enlargements (Figure 2C). Methodological details are now reported in the materials and methods section. We analyzed 30 randomly chosen images at 25.000x magnification. The results are representative of two independent experiments. We did not analyze autophagic vacuoles as they are seldom observed in ground cells by electron microscopy. They become more evident and numerous when autophagy is stimulated by stress (e.g. µG) or nutrient deprivation. However, this observation goes beyond the scope of this work but would certainly merit attention in future studies.
COMMENT 9
2.1.4. This is not a result that means anything. What ‘cell parameters’ that are ‘essentially stable’? This isn’t scientific. The reasoning for choosing 36h and not 24h is poor – if micrographs are being shown at 24h, surely results that match this time point should be used. A heatmap to show these miRNAs altered, perhaps even 1.5-fold, in a complete figure, would be useful. Further… how many replicates, cells, and so forth.
ANSWER 9
We thank the reviewer for the comments. Figure 2 has been modified by adding a representative TEM image of mitochondria at 36 hours of µG and the quantification of the altered mitochondria.
COMMENT 10
2.1.5. Enrichment analysis.
I do not understand, from this result, what is being shown. miRNAs affect multiple genes. But what has the list of other cancers to do with this? Figure 4 shows that all (or some? It isn’t stated). Of the genes are connected. No conclusions are drawn from this. And none in reference to NSCLC. What does enrichment, to produce Table 2, involve? Why are miRNAs that are altered (supp. Data), excluded?
ANSWER 10
We thank the reviewer for the comments. We have described this aspect in the paragraph 2.1.5 Enrichment analysis strategy. In addition, the representation of the molecular and biochemical relationships among the genes selected after the enrichment analysis is represented in figure 4.
COMMENT 11
Is this paper focussed on miRNA? In which case, what relevance is population doubling time or multinucleation?
ANSWER 11
Thank you to the referee for this comment. In this article we reported that exposure of A549 cells to µG is associated with several relevant morphological and physiological changes. In order to establish how these changes are related to the cell’s biochemistry and metabolism we undertook a differential analysis of microRNA expression in µG vs NG.
A paragraph was added in Results reporting that ‘MiRNAs modulated by µG are mainly involved in cell cycle regulation (P53, CDKN, E2F), apoptosis (BCL2, BIRC5), stress response (FOS, MAPK).
COMMENT 12
Discussion and conclusions drawn to not adequately explain any results found.
ANSWER 12
Thank you to the referee for this comment. We agree with the referee that a more detailed conclusion was necessary at this point. The discussion was then amended with the introduction of the following phrases (lines 222-236):
In this article we demonstrated that A549 cell exposure to µG is associated with several biochemical and molecular modifications linked to a series of pathological processes.
A relatively small number of genes (as highlighted in the heatmap, Fig. 3) appear responsible of the characteristic modifications in A549 exposed to µG.
These genes are closely related in central metabolic pathways where the phosphatidylinositol 3-kinase-Akt (PIK3R1-Akt) signaling pathway plays a crucial role (Jiang N, Dai Q, Su X, Fu J, Feng X, Peng J. Role of PI3K/AKT pathway in cancer: the framework of malignant behavior. Mol Biol Rep. 2020 Jun;47(6):4587-4629. doi: 10.1007/s11033-020-05435-1. Epub 2020 Apr 24.). PIK3R1-Akt is primarily regulated upstream with their negative regulator PTEN and downstream with a number of effectors including FOXO3. On another side these genes are linked with cell cycle control (CDKN2A/1A and RB) and with the transcription factors E2F1/3 and, finally with the control of the mitochondrial activity (BCL2). The control of all these highly interconnected paths appear under the control of TP53. The STRING representation (Fig. 4) of these relationships offers a picture of this complex pattern.
In conclusion These microRNAs and genes the relationships sketched in Fig. 4 may be used as tools in the experimental activity meant to define the µG biological effects.

Reviewer 2 Report
Authors present some data that could be worth being publish. However, they fail to provide a clear message what the meaning of these miRNA changes is.
Please state that this is simulated microgravity in the study
Abstract: please state clearer how the role of simulated micro-g is elucidated by this study.
Results, line 82: what do the authors mean with the statement that the [morphological parameters] “worsened protracting exposure to 72h”. What was bad was and what worsened?
Figure 1: please provide scale bars
Figure 2: please display scale bars bigger
Table 1: numbers displayed in the legend differ from those in the table. Please correct
Discussion
Line 170: where did author measure apoptosis?
Line 195-196: please explain this in more detail. Was is the significance of the miRNA alteration under simulated micro-g in your opinion?
Conclusion: line 300- 301 what does this statement mean? Please find a conclusion that sums up your results better.
Please use consistent abbreviations, not SM and MG alternating…
Author Response
Reviewer 2:
COMMENT 1
Authors present some data that could be worth being publish. However, they fail to provide a clear message what the meaning of these miRNA changes is.
Please state that this is simulated microgravity in the study
ANSWER 1
Thank you to the referee for this comment. Question 1 –The corrections have been made in the abstract and in the discussion. Question 2 – The corrections have been made in the abstract and in the discussion paragraph marked in yellow.
The simulated microgravity has been added in the title of the paper.
COMMENT 2
Abstract: please state clearer how the role of simulated micro-g is elucidated by this study
ANSWER 2
Thank you to the referee for this comment. The role of simulated micro-g has been elucidated in the abstract as requested.
COMMENT 3
Results, line 82: what do the authors mean with the statement that the [morphological parameters] “worsened protracting exposure to 72h”. What was bad was and what worsened?
ANSWER 3
We acknowledge the reviewer for the comments.Time point 72 h was deleted.
COMMENT 4
Figure 1: please provide scale bars.
ANSWER 4
We acknowledge the reviewer for the comments. The scale bars have been added in the Figure 1.
COMMENT 5
Figure 2: please display scale bars bigger
ANSWER 5
We acknowledge the reviewer for the comments. The scale bars have been corrected in the Figure 3. (ex Figure 2).
COMMENT 6
Table 1: numbers displayed in the legend differ from those in the table. Please correct
ANSWER 6
Table 1 was re-elaborated
COMMENT 7
Discussion Line 170: where did author measure apoptosis?
ANSWER 7
We acknowledge the reviewer for the comment. The apoptosis evaluation was added in the 2.1.1 Cell proliferation and cell cycle paragraph in lines 94-95., and in the Table 2 description line 192-193.
COMMENT 8
Line 195-196: please explain this in more detail. Was is the significance of the miRNA alteration under simulated micro-g in your opinion?
ANSWER 8
We acknowledge the reviewer for the comments. In discussion (lines 222-236) the following phrases were introduced to help to clarify this point:
In this article we demonstrated that A549 cell exposure to MG is associated with several biochemical and molecular modifications linked to a series of pathological processes.
A relatively small number of genes (as highlighted in the heatmap, Fig. 3) appear responsible of the characteristic modifications in A549 exposed to MG.
These genes are closely related in central metabolic pathways where the phosphatidylinositol 3-kinase-Akt (PIK3R1-Akt) signaling pathway plays a crucial role (Jiang N, Dai Q, Su X, Fu J, Feng X, Peng J. Role of PI3K/AKT pathway in cancer: the framework of malignant behavior. Mol Biol Rep. 2020 Jun;47(6):4587-4629. doi: 10.1007/s11033-020-05435-1. Epub 2020 Apr 24.). PIK3R1-Akt is primarily regulated upstream with their negative regulator PTEN and downstream with a number of effectors including FOXO3. On another side these genes are linked with cell cycle control (CDKN2A/1A and RB) and with the transcription factors E2F1/3 and, finally with the control of the mitochondrial activity (BCL2). The control of all these highly interconnected paths appear under the control of TP53. The STRING representation (Fig. 4) of these relationships offers a picture of this complex pattern.
In conclusion These microRNAs and genes the relationships sketched in Fig. 4 may be used as tools in the experimental activity meant to define the MG biological effects.
COMMENT 9
Conclusion: line 300- 301 what does this statement mean? Please find a conclusion that sums up your results better.
ANSWER 9
Thank you to the referee for this comment. The phrases reported on answer 8, in the reviewed Discussion, might be more appropriate to draw the conclusions of our study.
COMMENT 10
Please use consistent abbreviations, not SM and MG alternating.
ANSWER 10
Thank you to the referee for this comment. The acronym of microgravity on the text has been changed and corrected. We reported microgravity as µG.

Round 2
Reviewer 1 Report
I still do not understand the relevance of Figure 4 considering various cancers are listed in the figure but are not detailed anywhere else.
Check the PI3K (line 238) acronym - should it be PIK3R1 or PI3K?
for TEM, should experiments not be three replicates, not two (Line 297?), to get the mean?
Author Response
Dear Editor:
thank you for your letter stating that our manuscript entitles: “Microgravity effects on human adenocarcinoma alveolar epithelial cells. Characterization of morphological, functional, and epigenetic parameters”, should be acceptable for publication in International Journal of Molecular Sciences pending revisions.
Accordingly, we prepared a revised version of the manuscript acknowledging Referees’ and Editor’s comments as below specified:
Reviewer 1:
COMMENT 1 I still do not understand the relevance of Figure 4 considering various cancers are listed in the figure but are not detailed anywhere else.
ANSWER 1 We agree with this comment. Accordingly, Figure 4 has been deleted.
COMMENT 2 Check the PI3K (line 238) acronym - should it be PIK3R1 or PI3K?
ANSWER 2 The PIK3R gene is corrected it is reported in this form in the Gene Card database.
COMMENT 3: for TEM, should experiments not be three replicates, not two (Line 297?), to get the mean?
ANSWER 3 We have performed three technical replicates in two independent experiments.

Reviewer 2 Report
The work hast been considerably improved.
Two points are still to clarify:
Mitochondria react to micro-g for sure, but there is not enough evidence that they represent the major microgravity sensor inside the cell. Please rephrase your conclusion in a way that it is supported by your data.
Minor correction: p3, line 98: the phrase "worsened" is a judgemental term that is not appropriate in the results-section. Please find a more descriptive termi
Author Response
Reviewer 2:
COMMENT 1 Mitochondria react to micro-g for sure, but there is not enough evidence that they represent the major microgravity sensor inside the cell. Please rephrase your conclusion in a way that it is supported by your data.
ANSWER 1: In the Abstract conclusions have been rephrased and the statement that “Mitochondria represent the major microgravity sensor inside the cell”, deleted.
COMMENT 2 Minor correction: p3, line 98: the phrase "worsened" is a judgemental term that is not appropriate in the results-section. Please find a more descriptive termi
ANSWER 2: We agree with this comment. Accordingly, this sentence has been deleted.

Round 3
Reviewer 1 Report
no further comments. thank you.